# Numerical Analysis for Appropriate Positioning of Ferrous Wear Debris Sensors with Permanent Magnet in Gearbox Systems

**DOI:** 10.3390/s24030810

**Published:** 2024-01-26

**Authors:** Sung-Ho Hong

**Affiliations:** Department of Mechanical System Engineering, School of Creative Convergence Engineering, Dongguk University—WISE Campus, Gyeongju 38066, Republic of Korea; hongsh@dongguk.ac.kr; Tel.: +82-54-770-2211

**Keywords:** optimal position, gearbox, ferrous wear debris sensor, multi-physics, flow guide wall

## Abstract

In order to improve the measurement sensitivity of ferrous wear debris sensors with a permanent magnet, a new numerical approach to the appropriate position of the sensor is presented. Moreover, a flow guide wall is proposed as a way to concentrate flow around the ferrous particle sensors. The flow guide wall is intended to further improve measurement sensitivity by allowing the flow containing ferrous particles to flow around the sensor. Numerical analysis was performed using the multi-physics analysis method for the most representative gearbox of the sump-tank type. In condition diagnosis using ferrous wear debris sensors, the position of the sensor has a great influence. In other words, there are cases where no measurements occur, despite the presence of abnormal wear and damage due to the wrong sensor position. To determine the optimal sensor position, this study used flow analysis for the flow caused by the movement of the gear, electric and magnetic field analysis to implement the sensor, and a particle tracing technique to track particle trajectory. The new analysis method and results of this study will provide important information for selecting the optimal sensor location and for the effective application of ferrous wear debris sensors, and will contribute to the oil sensor-based condition diagnosis technology.

## 1. Introduction

Maintenance strategies are applied in terms of the reliability in various mechanical systems. Maintenance has been defined as the combination of technical and associated administrative actions intended to retain an item or system in, or restore it to, a state in which it can perform its required function (ISO 14224 [1]). The main purpose of maintenance is to reduce the adverse effects of breakdown and to increase availability at a low cost, in order to increase performance and improve the dependability level. That is, the key objective of maintenance management is “total asset life cycle optimization”, the maximization of the availability of a plant and/or equipment, and the reliability of these assets in order to achieve operational and/or business objectives [2,3]. In the field of machine condition diagnosis, condition diagnosis based on noise and vibration still occupies a large proportion. As with most mechanical devices, condition diagnosis technology based on vibration and signals is widely used to monitor gearboxes [4,5,6]. However, ferrous wear debris occur as a sign of gear abnormality, and with the development of lubricant sensors, condition diagnosis technology based on lubricant analysis is being widely applied. Therefore, condition diagnosis based on oil analysis occupies more than 20%, and the ratio is expected to increase in the near future due to the continued development of oil sensors and diagnostic algorithms [7].

Among oil sensors, the development of sensors for wear particles accounts for a large proportion, and among them, ferrous wear debris sensors are widely used for condition diagnosis because iron is widely used as a component of mechanical systems, and the generation of wear particles is a major indicator for abnormality diagnosis [8,9,10,11]. This study focused on the application of a ferrous wear debris sensor with a permanent magnet. This sensor can accumulate ferrous wear particles by use of a permanent magnet, so the sensor can not only measure the number of ferrous wear particles, but also prevent three-body abrasive wear. In previous research [10], different results have been obtained depending on the location of a ferrous wear debris sensor with a permanent magnet to diagnose the condition of an axle in construction equipment. Here, the axle performs a function similar to that of a gearbox. In one case, a ferrous wear particle was measured, but in the other case, even though severe wear and breakage occurred, the sensor did not measure it. In the latter case, the location of the sensor was far from the area where the wear occurred. Moreover, as a way to improve the sensitivity of this sensor, the internal design factors of the improvement were changed. By appropriately changing the shape of the core among various design factors, the maximum magnetic flux density was improved to more than twice that of the existing sensor [11].

Except for ferrous wear debris sensors with permanent magnets, most wear sensors mainly measure wear particles where a fine flow occurs. However, ferrous wear debris sensors combined with permanent magnets are applied to sump-type lubrication systems such as gearboxes and engines due to the advantage of measuring a large number of wear particles. As mentioned above, when applied to a system containing a large space or lubricating oil, such as a gearbox or an engine, the position of the sensor has a large effect on its sensitivity. The author believes that there are several methods for solving the problem [8]. The first is to improve sensor sensitivity by either changing the design of an existing sensor or developing a new type of sensor. However, in both cases, it is not an easy objective. There have been previous studies [12,13,14,15,16,17] to improve the sensitivity of oil sensors, but there is no study on the sensitivity improvement of ferrous wear debris sensors combined with permanent magnets. Another method to improve sensor sensitivity involves improvement in signal processing such as noise removal. The development of a new type of ferrous wear debris sensor has also been carried out [18,19,20]. In addition, there are improvements in sensor positioning to improve the sensitivity of the sensor. Most studies suggest methods to analyze or eliminate the causes of the sensor position errors [21,22,23,24,25,26,27,28,29,30,31]. Research has also been conducted to improve technology for analyzing wear particles. An improved ant colony algorithm [32], synchronized sampling [33], and deep learning with stochastic global optimization [34] are also used to effectively analyze wear particles. Most wear particle sensors measure in a small oil passage, but the ferrous wear debris sensor with a permanent magnet measures particles in a large volume of lubricant such as a gearbox and engines, as shown in Figure 1. Therefore, this ferrous wear debris sensor used for condition diagnosis in a relatively wide space has very different measurement results depending on the location, as shown in Figure 2. The ferrous wear debris sensor installed on the axle of construction equipment, which performs a similar role to the gearbox, did not detect abnormalities even though the gears of the axle were severely damaged. That is, it is necessary to select an appropriate sensor location in order to effectively perform condition diagnosis in sump-type lubrication systems such as a gearbox. Therefore, this study focused on how to select the optimal location of the sensor to improve its sensitivity. A numerical approach is suggested for selecting the optimal location of the ferrous wear debris sensor with a permanent magnet, considering the flow and the sensor’s measurement capability. Furthermore, this study tried to improve sensitivity by proposing a guide wall to generate the flow around the sensor.

## 2. Numerical Model and Methods

The ferrous wear debris sensor uses a permanent magnet to attach ferrous wear particles to the sensor and measures the number of ferrous wear particles using a change in the magnetic field. In addition, condition diagnosis of sump-type lubrications such as gearboxes and engines is mainly applied. Therefore, to analyze the optimal positioning of the sensor, sensitivity must be measured by installing the sensor in a control volume that simulates the gearbox. Figure 3 shows the numerical model for the ferrous wear debris sensor and gearbox. The cross-sectional shapes of the ferrous wear debris sensor are shown in Figure 3a. The geometries of the sensor are presented in Table 1. The sensor was placed in three positions while the position of the gears was fixed, as shown in Figure 3b. Sensors can be placed in various locations in sump-type lubrication systems such as gearboxes. In this study, the three most representative cases were selected and analyzed. However, it is difficult to determine the optimal sensor location only based on the analysis results for the three cases. Nevertheless, through this study, the sensitivity of the sensor can be numerically shown according to the locations of the three representative sensors and the importance of sensor location selection is intended to be demonstrated. In position-1, the sensor was installed on the floor, slightly away from the gears, and in position-2, the sensor was installed at the bottom, below the gears. In position-3, the sensor was installed on the right side wall, somewhat away from the gears. The actual gearbox is composed of various gears; however, for the convenience of analysis, only two gears are used to describe the main flow. Figure 3c shows the meshes of the gearbox with the sensor for the position-1 case. The total number of elements for the three position cases ranges from 2,995,456 to 3,074,755. In addition, a dense mesh is applied around the sensor and gears to ensure numerical accuracy. There is no difficulty in using the three analysis modules used in the numerical analysis, but trial and error is necessary to improve convergence when applying meshes between gears. The sensor is composed of a core, permanent magnet and case. The magnetic core is made of low-carbon steel M-50, and the B-H curve is the same as the value used in previous studies [11].

This analysis was performed on a closed system. For the gear, a Frozen Rotor study analysis was performed using the rotating domain function. As a boundary condition, an insulation condition was given on the symmetry plane. And infinite domains were applied to the outer boundary area surrounding the fluid. Even though a plane symmetry analysis was performed, it took about 2 days of calculation time per case using a computer with 3.00 GHz, 16 Core CPU, and 500 GB RAM specifications. 

The flow in the gearbox was laminar and the working conditions for numerical analyses are shown in Table 2. The particles used in the analysis were spherical and composed of iron with a density of 7800 kg/m^3^. The ferrous particles were sprinkled from above, in the midsection of the gears to simulate the occurrence of wear particles on the gears. During the initial two seconds of calculation, particles were injected at intervals of 0.05 s from the injection area. The total number of injected particles was 450. The viscosity and density of the lubricating oil were 0.04 Pa·s and 870 kg/m^3^, respectively. The number of teeth in the gear was 20, with a rotational speed of 1000 rpm. In addition, the diameter of the gears was 25 mm and the width was 12.5 mm.

In this research, a method of analyzing multiple physics was employed, and numerical calculations were performed using the commercial software COMSOL 6.0. The numerical analysis involved using an interface model for the electromagnetic field, a module for tracing particles, and the Navier–Stokes equation. Figure 4 shows the configuration of the numerical analysis modeling used in this analysis. The analysis proceeds sequentially from step 1 to step 3. Step 1 and step 2 are analyzed independently of each other. Step 1 uses the computational fluid dynamics (CFD) module to solve the velocity and pressure of the fluid, and step 2 uses the AC/DC module to solve the magnetic field potential. Finally, in step 3, the particle tracing module is used. This module calculates the velocity and positions of the particles using the drag force (*F_D_*) of the fluid obtained in step 1 and the magnetophoretic force (*F_ext_*) obtained in step 2. Step 1 and step 2 analyze the steady-state; however, in step 3, analysis of unsteady-state is performed. A detailed description of the governing equations is as follows. The COMSOL AC/DC module included the interface model for the electromagnetic field, which was utilized to determine the magnetic flux density of the ferrous wear debris sensor. Meanwhile, the particle tracing module enabled the calculation of the paths of individual particles by solving their equations of motion over time, which resulted in several different trajectories. This module made it possible to verify whether the ferrous wear debris sensor captured any particles and to analyze the trajectories of ferrous particles in the flow.

Equations (1)–(3) are used to calculate the sensor’s magnetic field based on Maxwell’s equations in the case of a magnetic field and no current condition. If the electromagnetic field and currents change gradually, the displacement current that is induced can be disregarded. It is referred to as the quasistatic approximation, a method often applied in low-frequency electromagnetic modeling, particularly when the structure’s size is considerably smaller than the wavelength.
(1)∇⋅B=0
(2)$$B=\mu_{o}\mu_{r}H$$
(3)$$H=-\nabla V_{m}$$
where *B*, *μ*_0_, *μ_r_*, *H*, and *V_m_* represent the magnetic flux intensity [T], permeability of vacuum [N/A^2^], relative permeability, magnetic field intensity [A/m], and magnetic scalar potential [A], respectively. In this program, the AC/DC module incorporates the electromagnetic field interface model, which computes the magnetic flux of the ferrous wear debris sensor. The numerical analysis involved employing the Navier–Stokes equations, the electromagnetic field interface model, and the particle tracing module. Within the AC/DC module, the electromagnetic field interface model (Equations (1)–(3)) is utilized to determine the magnetic flux of the ferrous wear debris sensor. Additionally, the applicable Navier–Stokes equations for rotating domains are expressed in Equations (4) and (5) [11].
(4)∇⋅(ρv)=0
(5)ρ(v⋅∇)v+2ρ Ω×v=∇⋅[−pI+τ]+F−ρ(Ω×(Ω×r))
where *ρ*, *Ω*, *I*, *τ*, and *F* are the density of fluid [kg/m^3^], angular velocity [rad/s], identity matrix, shear stress [Pa], and volume force [N/m^3^], respectively. Moreover, *v*, *r*, and *p* are the velocity [m/s], position vector [m], and pressure [Pa], respectively. 

The particle tracing module is utilized to compute the individual particles’ paths by solving their equations of motion over time, which allows for the evaluation of discrete trajectories.
(6)$$\begin{array}{l} \;\;\;\;\;\;\;\;\;\;\;\;\;\;\;\;\;\;\;\;\;\;\;\frac{d}{dt}(m_{p}v_{1})=F_{D}+F_{ext}\\ F_{D}=\frac{1}{\tau_{p}}m_{p}(v-v_{1}),\;\;\;F_{ext}=2\pi r_{p}^{3}\mu_{o\;}\mu_{r\;}K{\left|H\right|}^{2},\;\;\;\tau_{p}=\frac{\rho_{p}d_{p}^{2}}{18\mu }\; \end{array}$$
where, *m_p_*, *v*_1_, *F_D_*, and *F_ext_* express the particle mass [kg], velocity vector of the particle [m/s], drag force [N], and magnetophoretic force [N]. In addition, *τ_p_*, *d_p_*, *r_p_*, *ρ_p_*, *μ* and *K* mean particle velocity response time [s], particle diameter [m], particle radius [m], particle density [kg/m^3^], dynamic viscosity of fluid [Pa∙s], and nondimensional parameters. The motion of particles in a fluid follows Newton’s second law, which states that the net force on an object is equal to the time derivative of its linear momentum in an inertial reference frame, as shown in Equation (6) [11].

An infinite boundary condition was set for the outermost boundary. In this case, the magnetic vector continuously exits the outermost layer. Therefore, if it is explained as a boundary condition, it can be given meaning as a continuous condition. Moreover, the fluid velocity on all walls except the rotating gear wall was set to 0. In other words, no-slip boundary conditions were applied in most cases. A stick condition of zero velocity was applied to the wall of the sensor in the plane symmetry plane. In addition, in the symmetry plane, a magnetic insulation boundary condition (n∙*B* = 0) was set.

## 3. Numerical Results and Discussions

### 3.1. Sensitivity Evaluation According to the Sensor Position

The sensitivity of the sensor was evaluated by changing the position of the sensor in a situation where the position and working condition of the gears was not changed. The ferrous wear debris sensor uses a permanent magnet to attach ferrous wear particles in the lubricant to the sensor and measures the number of particles through inductance and magnetic field change by the attached ferrous wear particles. In the ferrous wear debris sensor, the number of particles attached to the sensor is closely related to its sensitivity. Therefore, the sensitivity of the ferrous wear debris sensor was evaluated by the number of ferrous particles attached to the sensor. The trajectories, due to the flow caused by the movement of the gears, of 450 ferrous particles injected between the gears are shown in Figure 5. The trajectories of the particles at the three positions of the sensor are slightly different; however, there is no change in the number of collected particles after 18 s. Figure 6 shows the distribution of magnetic flux density for the three positions. The maximum magnetic flux density is formed at the top of the sensor and its magnitude is 0.471 T (Tesla). Figure 7 shows the distribution of the magnetic force lines around the sensor, and the particles attached to the sensor, when the sensor is at position-1. As shown in the picture at the right of Figure 7, the sensitivity of the sensor was evaluated by identifying the number of particles attached to the sensor.

Figure 8 shows the number of ferrous particles attached to the sensor over time. In position-1 and position-3, the number of ferrous particles collected did not change to five after about 15 s. In position-2, the number of ferrous particles collected is constant at 56 after about 7 s. For position-2, the number of ferrous particles collected increased by 1020% compared to position-1 and position-3. This is because the location of the sensor is close to the point where ferrous wear particles of the gears are generated and also close to the place where the main flow, due to the movement of the gear, is generated when the sensor is installed at position-2. Moreover, in the case of position-2, the time during which the number of collected particles did not change is short. This means that anomalies can be detected earlier than other sensor positions. That is, it is more effective in diagnosing abnormalities when the sensor is in position-2. Therefore, it is important to select the location of the sensor in terms of the early detection of abnormal symptoms. The reason is that the time of diagnosis is closely related to the degree of damage to the machine. As mentioned above, the sensitivity of the sensor and the time taken to detect anomalies are correlated with the main flow and the location of the sensor. Figure 9 shows streamlines in the gearbox according to the three positions of the sensor. When the sensor is in position-2, it can be seen that the main flow due to the movement of the gear meets the top of the sensor. The result explains the improved sensitivity when the sensor is in position-2.

The actual gearbox has more gears and a more complex structure than the model used in the analysis. However, the direction of the main flow due to the movement of the gears can be sufficiently investigated through CFD analysis. Therefore, in order to effectively use the ferrous wear debris sensor for condition diagnosis in sump-type lubrication systems, it is necessary to select a position where measurement is the easiest by examining the main flow. In other words, the sensor must be selected in a location where the main flow is developed in the lubrication system, where there is sufficient space and where installation is easy. If the main flow is developed, but the distance between the sensor and the gear is very close, it may not be suitable in a place where the sensor may break down or installation is difficult.

### 3.2. Improvement of Sensitivity Using a Flow Guide Wall

Due to the large size of the gears in the gearbox, it is sometimes difficult to install the sensor in the place where the main flow occurs because the distance to the sensor is short and damage may occur. Thus, if the sensor needs to be placed at a location away from the main flow, such as position-1 and position-3, a method to compensate for this is needed. As a complementary method, a flow guide wall was applied that allows a part of the main flow to proceed around the sensor which is installed away from the main flow. As an example, two types of flow guide walls were applied for the case where the sensor was located at position-1, as shown in Figure 10. Position-1+guide-1 is the case where a part of the main flow proceeds toward the sensor using the flow guide wall. In position-1+guide-2, the returning flow colliding from the right wall flows towards the sensor using the flow guide wall. Moreover, guide-1 is designed to narrow in the direction toward the sensor to collect the flow. The streamlines within the gearbox are shown in Figure 10, where the two types of flow guide walls are applied. When guide-1 is applied, a large part of the main flow is directed toward the sensor, and when guide-2 is applied, a portion of the returning flow from the wall is directed toward the sensor. That is, it is confirmed through the streamlines that the proposed flow guide walls could induce the flow toward the sensor. In terms of sensor sensitivity, a comparison of the results with and without the flow guide walls is shown in Figure 11. When guide-1 was applied, the number of collected particles increased by 620% compared to where it was not applied. Moreover, when guide-2 was applied, this number increased by 300% compared to where the flow guide wall was not applied. Figure 12 shows the trajectories of ferrous particles for three cases at 18 s. In three cases, the sensor is located at position-1 and there is no guide wall case, guide-1 case, or guide-2 case. As shown in the red dotted line in Figure 12, it can be seen that more particle trajectories are formed toward the sensor when there is a flow guide wall than when there is no flow guide wall. This particle trajectory shows that the flow guide walls improve the sensitivity of the sensor. In addition, it was confirmed that guide-1 can collect particles more effectively than guide-2 by directing the main flow generated from the gear toward the sensor.

The application of guide-1 is more effective in improving the sensitivity of the sensor because it allows a higher proportion of the main flow to go toward the sensor. The sensitivity of the sensor using the flow guide wall is lower than that of the sensor at position-2, which is installed in the main flow. Through the results of this study, it was confirmed that the optimal sensor position is the location where it directly meets the main flow, and if the sensor is installed at a location away from the main flow due to a sensor installation problem, sensitivity can be supplemented by applying the flow guide wall. For each gearbox, the shape of the gears or the direction of the main flow is different. Therefore, to effectively perform condition diagnosis using a ferrous wear debris sensor in a sump-type lubrication system such as gearboxes and engines, it is necessary to determine the optimal location of the sensor through flow analysis. Next, through flow analysis, the shape of the guide is determined so that the main flow can be directed toward the sensor. Moreover, the width and length of the guide must be designed properly so that it does not cause collision with gears and is easy to manufacture or install.

## 4. Conclusions

In this study, a numerical analysis method for selecting the optimal sensor position was suggested for the first time as a way to effectively diagnose machine conditions using a ferrous wear debris sensor with a permanent magnet in lubrication systems. The ferrous wear debris sensor is widely used for condition diagnosis in sump-type lubrication systems such as engines and gear-boxes where iron-based wear particles are mainly generated. However, if the sensor is installed in an inappropriate location, it may fail to diagnose abnormalities. To solve this problem, the sensitivity of the sensor was evaluated depending on the position in a lubrication system such as a gearbox. The sensitivity of the sensor was defined as the number of ferrous particles attached to the sensor because this sensor attached ferrous particles to the sensor with a permanent magnet and measured the number of ferrous particles through changes in inductance. By analyzing the streamlines of the flow field, the location with good sensor sensitivity was where the main flow flows in the lubrication system. Therefore, in order to increase sensitivity, the sensor should be installed at a location where it directly meets the main flow. To determine the optimal location with high sensitivity, it is necessary to investigate the main flow through flow analysis. In addition, when the sensor is installed away from the main flow due to installation problems, sensor sensitivity can be improved by utilizing the flow guide wall which directs a large part of the main flow towards the sensor. Therefore, flow guide walls are another way to improve the sensitivity of the sensor and should be designed based on flow analysis.

## Figures and Tables

**Figure 1 sensors-24-00810-f001:**
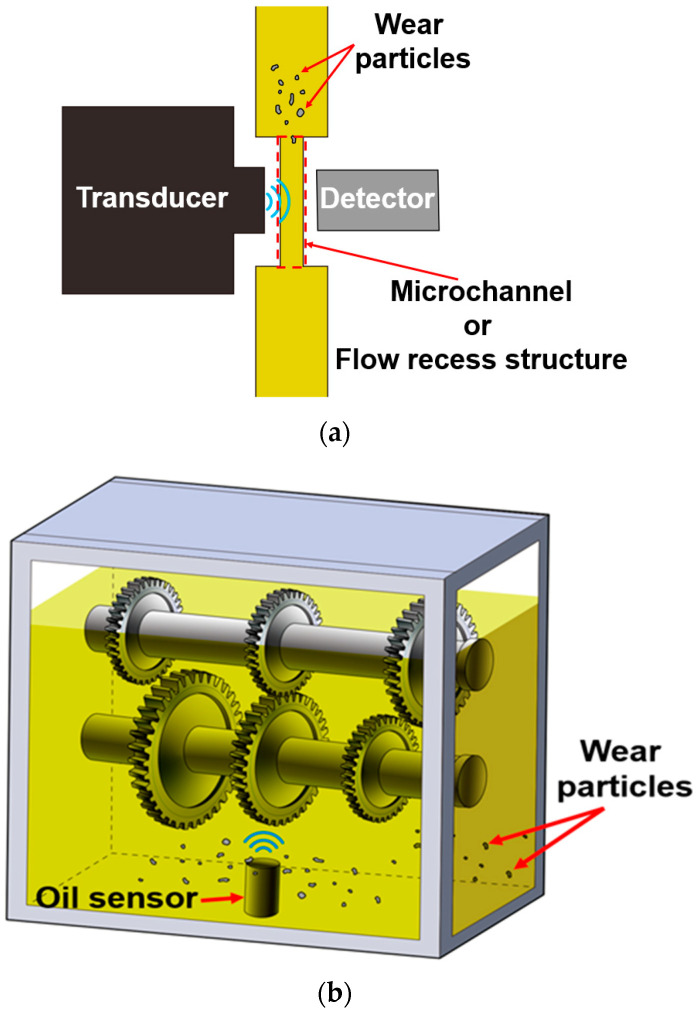
Representative methods for measuring wear particles through an oil sensor: (**a**) Measurements in the flow of microchannels; (**b**) Bulk measurement in a large space such as the gearbox.

**Figure 2 sensors-24-00810-f002:**
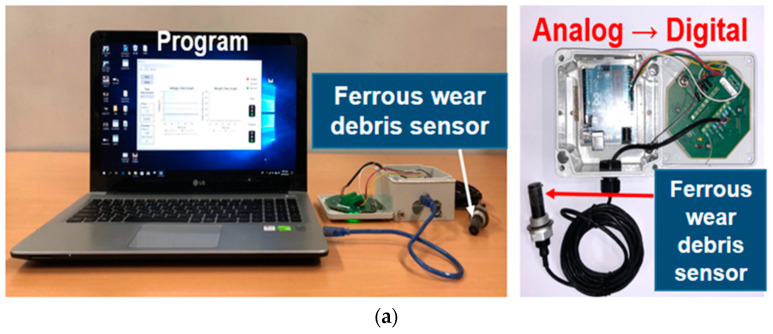
Experimental results using a ferrous wear debris sensor with a permanent magnet in the axle: (**a**) Condition monitoring program and system; (**b**) Case result where the gear damage was not measured by the sensor and broken gears inside the axle [11].

**Figure 3 sensors-24-00810-f003:**
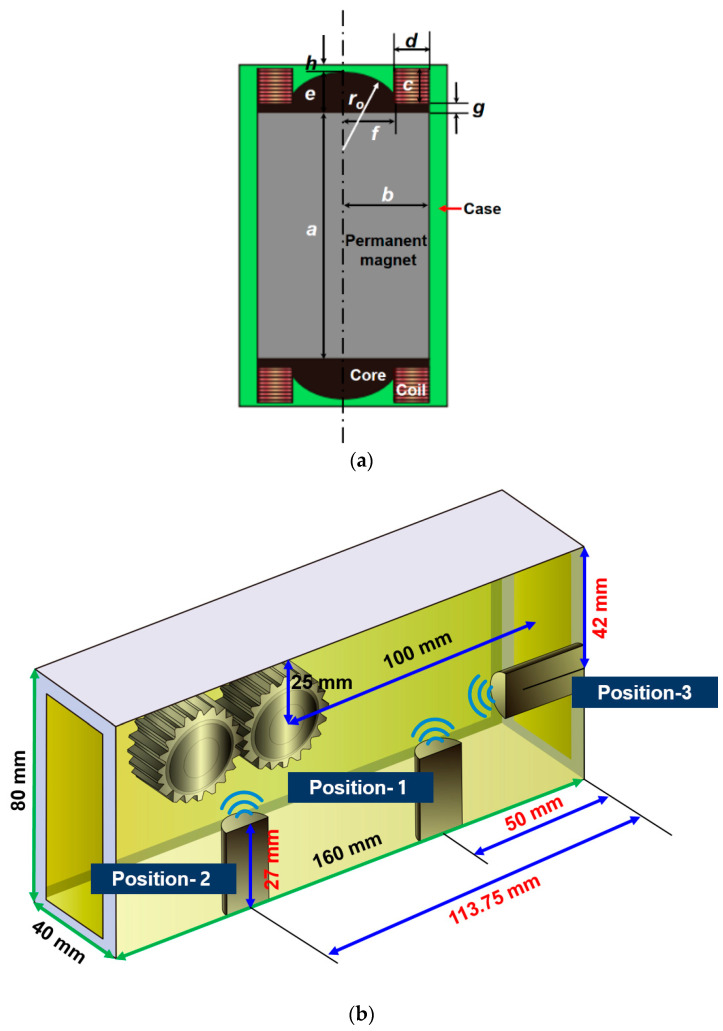
Numerical model for a gearbox with a ferrous wear debris sensor: (**a**) Geometries of the ferrous wear debris sensor; (**b**) Geometries of the gearbox with a ferrous wear debris sensor; (**c**) Mesh of the gearbox for the position-1 case.

**Figure 4 sensors-24-00810-f004:**
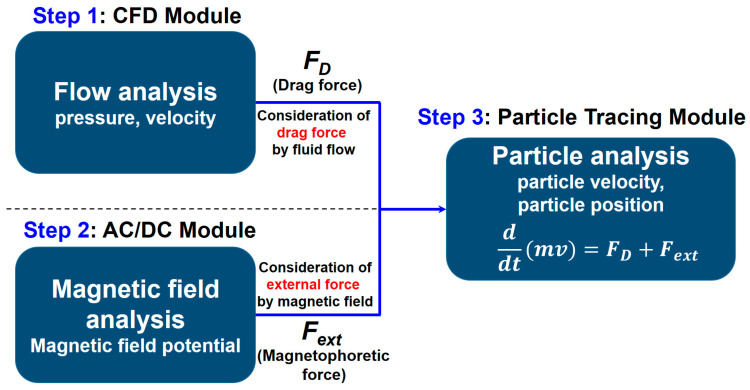
Composition of numerical modeling.

**Figure 5 sensors-24-00810-f005:**
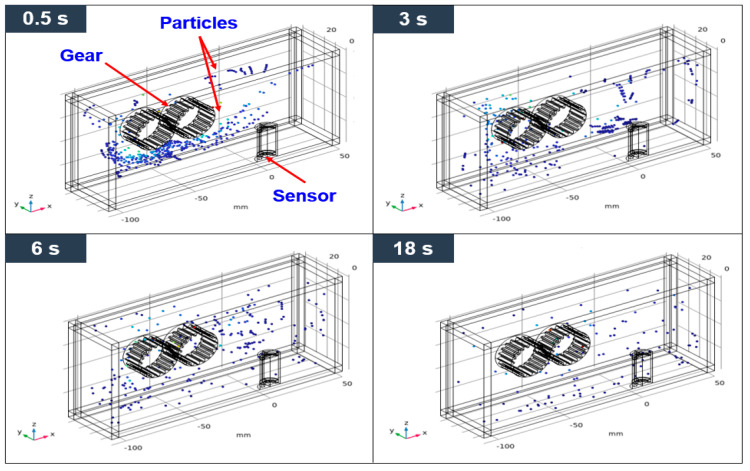
Particle trajectories for position-1 with time variations.

**Figure 6 sensors-24-00810-f006:**
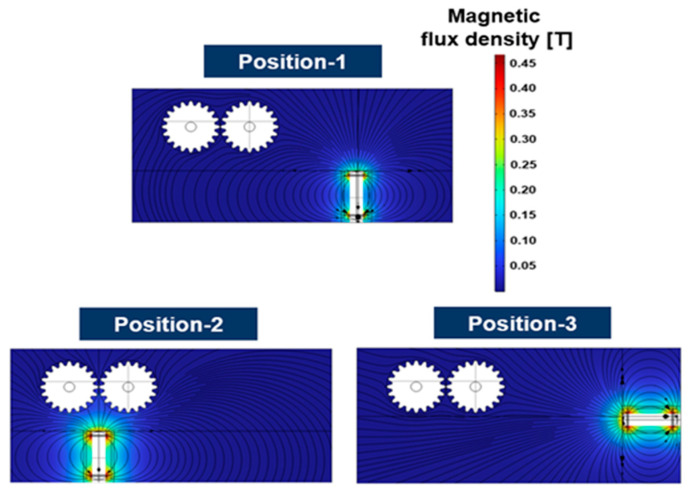
Magnetic flux density for the three different positions.

**Figure 7 sensors-24-00810-f007:**
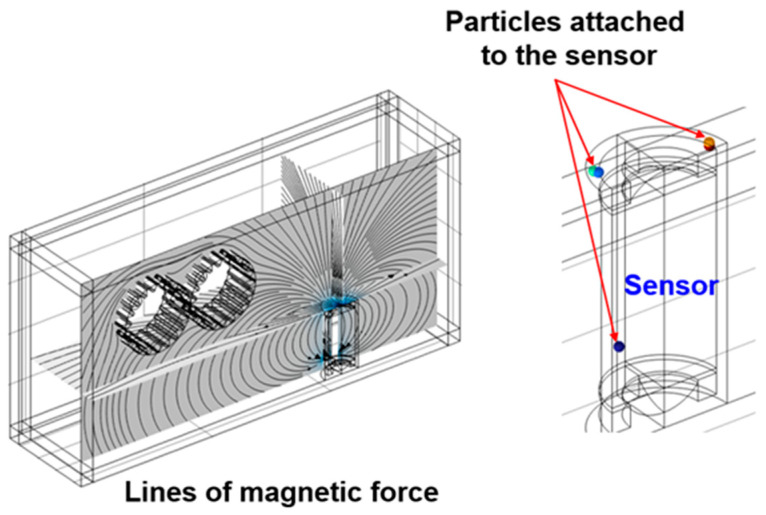
Lines of magnetic force and particles attached to the sensor for position-1.

**Figure 8 sensors-24-00810-f008:**
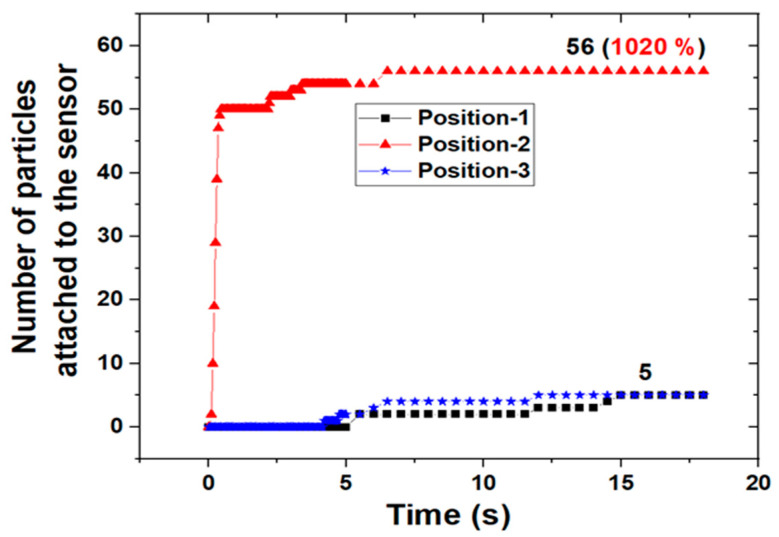
Number of particles attached to the sensor for the three positions with time variations.

**Figure 9 sensors-24-00810-f009:**
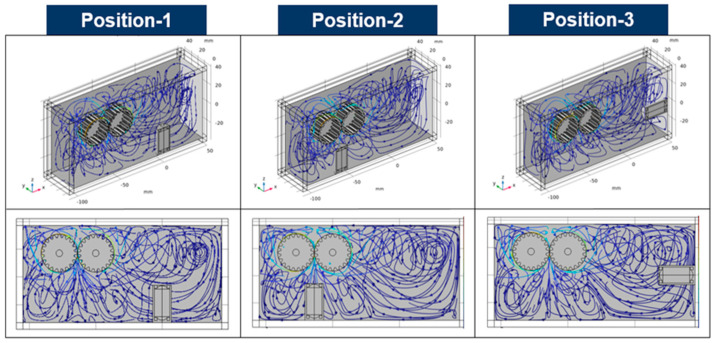
Streamlines for the three positions.

**Figure 10 sensors-24-00810-f010:**
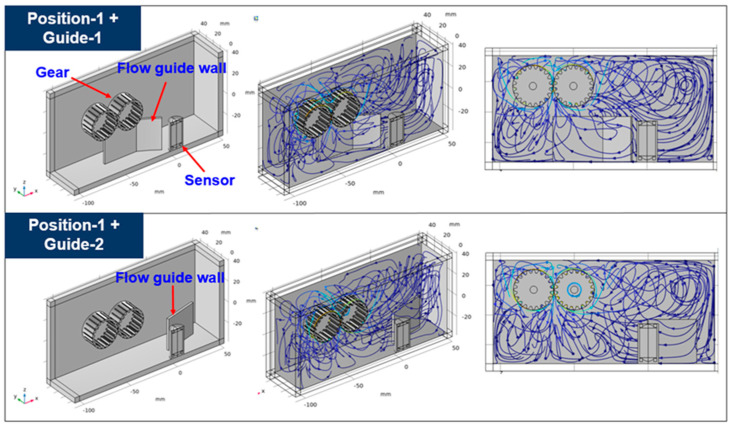
Streamlines for position-1 case with two types of flow guide wall.

**Figure 11 sensors-24-00810-f011:**
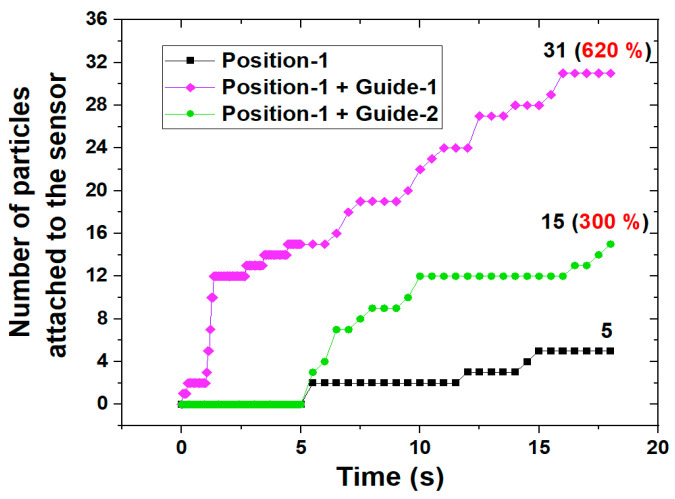
Number of particles attached to the sensor according to the application or absence of a flow guide wall.

**Figure 12 sensors-24-00810-f012:**
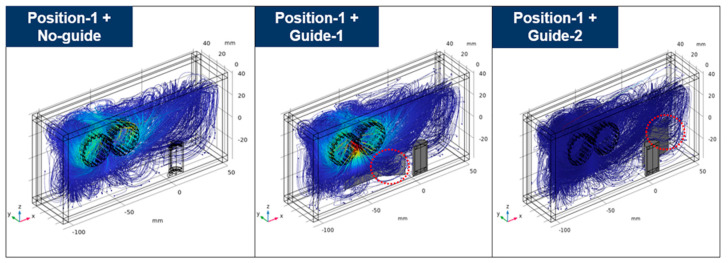
Particle trajectories at 18 s according to the application or absence of a flow guide wall.

**Table 1 sensors-24-00810-t001:** Geometries of the ferrous wear debris sensor with a permanent magnet.

Parameters	Values	Parameters	Values
*a* [mm]	20	*f* [mm]	3.75
*b* [mm]	5.8	*g* [mm]	0.5
*c* [mm]	2.2	*h* [mm]	0.4
*d* [mm]	2.05	*r*_1_ [mm]	5
*e* [mm]	2.5		

**Table 2 sensors-24-00810-t002:** Working conditions for numerical analysis.

Items	Conditions	Items	Conditions
Particle diameter	10 μm	Number of particles	450
Particle material	Steel	Particle shape	Sphere
Particle density	7800 kg/m^3^	Relative permeability of the particle	1000
Dynamic viscosity of the lubricant	0.04 Pa·s	Density of the lubricant	870 kg/m^3^
Number of teeth in the gear	20	Rotational speed of the gears	1000 rpm

## Data Availability

Data are contained within the article.

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
