# Peer review of "Numerical Analysis for Appropriate Positioning of Ferrous Wear Debris Sensors with Permanent Magnet in Gearbox Systems"

_sensors, 2024, doi:10.3390/s24030810_

Round 1
Reviewer 1 Report
Comments and Suggestions for Authors
Monitoring wear particles in the gear ratios of machine components and assemblies is very important today. There are also known methods of monitoring with partial changes in the design of the housing and the use of various kinds of counter-indicators for online monitoring of the condition of rubbing pairs. The study should have paid more attention to the analysis of already known methods and methods for monitoring wear marks of rubbing pairs at the global level. Accordingly, this would be reflected in the list of references used
Author Response
Dear editor and reviewer
I am attaching the reviewer's response as a file.
Sincerely yours,
Sung-Ho Hong

Reviewer 2 Report
Comments and Suggestions for Authors
Here are some suggestions:
1. Please enrich your content and add further results or discussion. Please add detailed comparisons of results before and after using the flow guide wall. The article only performs simulation analysis, which makes it difficult to determine the reliability in practical applications. It is recommended to add some experiments.
2. The analysis is limited to three positions so that the generalizability of this research is not reflected. It is desirable to obtain generalized laws about optimal positioning of sensors developed in this research.
3. There is a shortage of references. Please look for more scientific papers and describe previous research more completely. This would help you to show the innovation and necessity of your research better.
4. The conclusion should be improved to emphasize the contribution of this research. It would be better if the results were explained more clearly.
5. What are the limitations of this research? What is the next research direction?
6. Please check the language carefully. For example, in line 203 equation (6) is missing and the first sentence in line 326 is not understandable.
Comments on the Quality of English LanguageModerate editing of English language required.
Author Response
Dear editor and reviewer
I am attaching the reviewer's response as a file. Because there is a picture file in the answer.
Sincerely yours,
Sung-Ho Hong

Reviewer 3 Report
Comments and Suggestions for Authors
When gear particles enter the meshing area, they may lead to gear wear and failure, thus this study is of great significance. The author's ideas are also interesting. I think there are still some questions to consider:
1 Why do the particles attached to the sensor not change after a period of time? Where are the other particles?
2 What is the principle of guide wall design? The author needs to give a clear explanation.
3 How many particles in the oil can determine the failure of the oil? Does such a standard exist? In the initial stage of gearbox operation, running-in wear is inevitable due to manufacturing and assembly errors, thus the existence of wear particles in oil is also almost inevitable.
4 Iron gear particles enter the meshing area, which may lead to gear wear and failure. Although the guide wall can improve the sensitivity of the sensor, it is also an important question whether it is possible to cause more particles to enter the meshing area.
Author Response

(The authors gave the same response as above.)

Round 2
Reviewer 2 Report
Comments and Suggestions for Authors
No comments
Author Response
Thanks
Reviewer 3 Report
Comments and Suggestions for Authors
All my comments are well responsed. I think it can be accepted.
Author Response
Thanks